# Therapeutic Effects of Stem Cells from Human Exfoliated Deciduous Teeth and Their Conditioned Medium in Cisplatin-Induced Acute Kidney Injury: An In Vivo Study

**DOI:** 10.3390/biology14091305

**Published:** 2025-09-22

**Authors:** Ardeshir Talebi, Sahar Talebi, Sara Nikpoor, Nosrat Nourbakhsh, Hossein Talebi, Sareh Soroushzadeh, Miguel Gómez-Polo, Seyed Ali Mosaddad

**Affiliations:** 1Department of Pathology, Medical School, Isfahan University of Medical Sciences, Isfahan 81746-73461, Iran; dr.sara.nikpoor@gmail.com; 2Water and Electrolyte Research Center, Isfahan University of Medical Sciences, Isfahan 81746-73461, Iran; 3Dental Research Center, School of Dentistry, Isfahan University of Medical Sciences, Isfahan 81746-73461, Iran; sahar.talebi.dds@gmail.com; 4Department of Pediatric Dentistry, School of Dentistry, Isfahan University of Medical Sciences, Isfahan 81746-73461, Iran; nourbakhsh@dnt.mui.ac.ir; 5Department of Cellular and Molecular Biology, Falavarjan Islamic Azad University, Isfahan 84517-31167, Iran; hossein.talebi.cmb@gmail.com; 6Iran Academic Center for Education, Culture and Research, Isfahan Branch, Isfahan 39998-81551, Iran; sarasorooshzadeh@gmail.com; 7Department of Conservative Dentistry and Prosthodontics, School of Dentistry, Complutense University of Madrid, 28040 Madrid, Spain; semosadd@ucm.es; 8Department of Research Analytics, Saveetha Institute of Medical and Technical Sciences, Saveetha Dental College and Hospitals, Saveetha University, Chennai 600077, India

**Keywords:** acute kidney injury, mesenchymal stem cells, conditioned culture media, cisplatin, extracellular vesicles

## Abstract

This study tested whether stem cells from baby teeth or the liquid they release could reduce kidney damage caused by chemotherapy. In rats, both treatments improved kidney function, but the cell-free liquid was more effective. The results suggest that the healing effect comes mainly from substances released by the cells, offering a safer and simpler treatment option.

## 1. Introduction

The kidney is a vital organ responsible for detoxification, extracellular fluid regulation, homeostasis, and the excretion of metabolic waste products [1]. Acute kidney injury (AKI) is a prevalent and potentially life-threatening condition, particularly among hospitalized patients, with an incidence rate of 10–15% in general hospital admissions and approximately 50% in intensive care unit (ICU) patients [2]. AKI is characterized by a rapid decline in kidney function due to various etiologies, including nephrotoxins, systemic diseases, and ischemic injury. Among nephrotoxins, drug-induced causes account for nearly 20% of hospital-acquired AKI cases [3,4,5].

Cisplatin, a platinum-based chemotherapeutic agent, is widely used to treat various solid tumors, including bladder, head and neck, ovarian, and lung cancers [6]. However, its clinical application is often limited by its nephrotoxic effects, leading to AKI. The kidney, serving as the primary excretory route for cisplatin, accumulates the drug in proximal tubular epithelial cells at concentrations approximately five times higher than serum levels [6]. This accumulation triggers oxidative stress, mitochondrial dysfunction, apoptosis, and inflammation, ultimately resulting in tubular necrosis and renal dysfunction. Due to these well-documented effects, cisplatin is commonly used to induce AKI in experimental animal models.

Given the high morbidity and mortality associated with AKI, effective therapeutic strategies are needed to mitigate kidney damage and enhance recovery [7]. Mesenchymal stem cells (MSCs) have gained attention as a promising treatment for AKI due to their regenerative and immunomodulatory properties [8,9,10]. Rehab H. Ashour et al. [11] investigated the effects of allogenic and xenogeneic mesenchymal stem cells (MSCs) on cisplatin-induced acute kidney injury (AKI) in Sprague-Dawley rats. Their results demonstrated that a single early intravenous (IV) administration of rat bone marrow-derived stem cells (rBMSCs), human adipose tissue-derived stem cells (hADSCs), or human amniotic fluid-derived stem cells (hAFSCs) provided a renoprotective effect against cisplatin-induced AKI by reducing oxidative stress markers. Comparable protective effects were also observed following intraperitoneal injection of MSC-conditioned medium (MSC-CM), supporting the notion that MSCs mediate their therapeutic effects primarily through paracrine mechanisms [12,13]. MSCs exert their therapeutic effects primarily through paracrine signaling by secreting trophic factors and extracellular vesicles (EVs), including exosomes (30–150 nm), microvesicles (200–1000 nm), and apoptotic bodies (800–5000 nm) [14]. These EVs facilitate intercellular communication and have demonstrated potential in reducing oxidative stress, inflammation, and apoptosis in AKI models [12,13]. However, concerns regarding MSC viability, immune rejection, and potential tumorigenicity present significant challenges to their clinical application [15].

Stem cells from human exfoliated deciduous teeth (SHED) have recently been identified as a novel and potent source of MSCs. These cells exhibit high proliferative capacity, enhanced secretion of growth factors, and strong regenerative potential [16,17,18]. Unlike bone marrow-derived MSCs (BM-MSCs), SHED can be obtained through a non-invasive procedure, making them a more accessible alternative for therapeutic applications [16]. Preclinical studies have demonstrated their efficacy in various models, including bone regeneration, neural repair, and ischemic injuries [19,20,21,22,23]. However, their application in acute kidney injury (AKI) remains largely unexplored, particularly regarding their secretome-based effects.

The use of MSC-conditioned medium (MSC-CM) has emerged as an alternative to cell transplantation, as it contains extracellular vesicles (EVs) and bioactive factors responsible for the therapeutic benefits of MSCs. Studies have shown that MSC-CM can achieve similar or superior outcomes compared to direct MSC administration while avoiding risks such as immune rejection and tumorigenicity [13,24,25]. Among MSC sources, SHED may offer enhanced clinical utility due to their accessibility and secretory profile. Yet, the therapeutic potential of SHED-derived conditioned medium (SHED-CM) in cisplatin-induced AKI has not been previously investigated. This study aimed to compare the effects of SHED and SHED-CM in this model to determine whether a cell-free, SHED-CM-based approach could serve as a safer and equally effective strategy for AKI treatment.

## 2. Materials and Methods

### 2.1. Ethical Considerations

All experimental procedures were approved by the Isfahan University of Medical Sciences Animal Ethics Committee (IR.MUI.REC.1399.038) and adhered to ethical guidelines for the humane handling and care of laboratory animals.

### 2.2. Animal Model and Study Design

This interventional, controlled study was conducted using 21 male Wistar rats aged 6–7 weeks, weighing between 200–300 g, obtained from the Royan Biotechnology Research Institute, Isfahan, Iran. The animals were housed under standard laboratory conditions, including a temperature of 20–25 °C, a regular 12 h light–dark cycle, and unrestricted access to food and water. The sample size was determined based on a previous study [26], with seven rats assigned to each group.

To induce AKI, all rats received a single intraperitoneal injection of cisplatin (7.5 mg/kg; Mylan, Saint-Priest, France). The control group received no further intervention after cisplatin administration. The SHED group received three intraperitoneal injections of SHED (500,000 cells/1 mL per injection) on alternate days for one week, while the SHED-CM group received three intraperitoneal injections of SHED-conditioned medium (1 mL per injection) on the same schedule. All injections were administered one day after cisplatin injection. Rats were weighed daily for eight consecutive days, and urine samples were collected on the eighth day for biochemical analysis. At the end of the study, all animals were anesthetized with chloral hydrate (450 mg/kg; Solarbio, Beijing, China) and xylazine (5 mg/kg; Bioveta, Ivanovice na Hané, Czech Republic) before euthanasia. The kidneys were harvested for histopathological and biochemical assessments.

### 2.3. SHED Isolation and Culture

SHED cells were obtained from the Iranian Banking and Developing SHED Knowledge-Based Company (Dental Stem Cell Bank, Tehran, Iran). According to the supplier’s documentation, the SHED used in this study were pooled samples derived from three healthy donors (age range: 6–8 years) obtained under standardized collection protocols with parental informed consent. Additional donor demographic details, such as sex or individual clinical histories, were not provided due to anonymization policies. SHED were isolated by enzymatic digestion of dental pulp tissue from exfoliated deciduous teeth under sterile conditions, following the supplier’s standardized protocol. Briefly, pulp tissues were minced and digested with collagenase type I (3 mg/mL) and dispase (4 mg/mL) at 37 °C for 1 h to obtain single-cell suspensions. The resulting cells were cultured in Dulbecco’s Modified Eagle Medium (DMEM) supplemented with 10% fetal bovine serum (FBS) and antibiotics, with medium changes every three days. SHED identity and quality were confirmed by the supplier using flow cytometry analysis, showing positive expression of mesenchymal stem cell markers (CD73, CD90, CD105) and negative expression of hematopoietic markers (CD34, CD45). Multipotency was further validated via osteogenic and adipogenic differentiation assays. Three frozen vials, each containing one million SHED at passage 6, were thawed and cultured in DMEM (Sigma-Aldrich, St. Louis, MO, USA) supplemented with 10% fetal bovine serum (FBS; Gibco, Grand Island, NY, USA), 1% non-essential amino acids (NEAA; Gibco, Grand Island, NY, USA), and 1% GlutaMAX (Gibco, Grand Island, NY, USA). The cells were expanded in T75 culture flasks (SPL, Pocheon, South Korea) and maintained at 37 °C with 5% CO_2_ until reaching 80% confluency. Half of the cultured cells were used for direct SHED injections, while the remaining portion was cultured again for SHED-CM collection. The morphology of the cells was examined under an inverted microscope to identify mesenchymal stem cells.

### 2.4. Preparation of SHED-Conditioned Medium (SHED-CM)

To prepare SHED-CM, SHED were cultured at a density of 10^4^ cells/cm^2^ in T75 flasks (SPL, Korea). Once the cells reached approximately 80% confluency, the serum concentration in the culture medium was gradually reduced. After complete serum removal, the cells were washed with phosphate-buffered saline (PBS) and incubated with serum-free DMEM for 48 h. Following incubation, the conditioned medium was collected, centrifuged at 3000× *g* for 10 min to remove cellular debris, and filtered through a 0.2-µm membrane filter (Corning, Corning, NY, USA). The collected SHED-CM was aliquoted and stored at −80 °C until further use.

### 2.5. Protein Assay and Structural Evaluation of SHED-CM

The protein concentration of SHED-CM was measured using a bicinchoninic acid (BCA) protein quantification kit (Pars Tous, Mashhad, Iran). SHED-CM samples were lysed with radioimmune precipitation assay (RIPA) buffer containing protease and phosphatase inhibitors and subjected to ultrasonic disruption for 30 min in an ice bath. The presence of extracellular vesicles (EVs) in SHED-CM was confirmed by scanning electron microscopy (SEM). Samples were fixed, dehydrated, coated with gold, and analyzed under a secondary electron (SE) mode at 15 kV with a working distance of 10 mm.

### 2.6. Renal Function and Biochemical Analyses

Renal function was evaluated by measuring urine volume and urinary protein concentration. A 6 h urine sample was collected on the eighth day using a metabolic cage, and urine volume was measured using a 2 mL syringe. Urinary protein concentration was assessed using a Mindray Autoanalyzer (Mindray, Shenzhen, China), with values expressed in mg/dL. Kidney weight was recorded immediately after dissection and expressed in grams.

### 2.7. Histological Assessment of Kidney Tissue

Excised kidneys were fixed in 10% formalin (Pars Chemie, Tehran, Iran), embedded in paraffin, and sectioned into 4-µm-thick slices. Tissue sections were stained with hematoxylin and eosin (H&E) and examined under a light microscope by a pathologist blinded to the study groups. The severity of kidney injury was scored based on a tubular injury scoring system, which categorized the percentage of tubule destruction into five grades [27,28]: score 0 for normal histology, score 1 for ≤24% injury, score 2 for 25–49% injury, score 3 for 50–74% injury, and score 4 for 75–100% injury. A minimum of 10 randomly selected fields at 100× magnification were assessed for each sample. Representative Grade 0 histological images, used for comparison and as a reference in the tubular injury scoring system, were obtained from healthy Wistar rats analyzed in previous experiments performed by the same research team under identical histological processing and imaging conditions.

### 2.8. Statistical Analysis

All data were analyzed using SPSS version 22 (IBM Corp., Armonk, NY, USA). Data were initially tested for normality using the Shapiro–Wilk test and for homogeneity of variances using Levene’s test. If both assumptions were met, comparisons between groups were performed using one-way analysis of variance (ANOVA) followed by Tukey’s post hoc test. For variables that did not meet normality or homogeneity assumptions, the Kruskal–Wallis test followed by Dunn’s post hoc test with Bonferroni correction was applied. Results are presented as mean ± standard deviation (SD). A *p*-value of <0.05 was considered statistically significant.

## 3. Results

### 3.1. Characterization of SHED

The morphology of SHED was examined using an inverted microscope, revealing a homogeneous population of spindle-shaped fibroblast-like cells 48 h after the first passage (Figure 1). These cells displayed typical mesenchymal stem cell morphology, confirming their successful isolation and culture.

### 3.2. SHED-CM Protein Concentration and Structural Evaluation

The protein concentration of SHED-CM was determined to be 527 μg/mL using a BCA protein assay. SEM images demonstrated the presence of extracellular vesicles within SHED-CM, with sizes ranging from 40 nm to 1 μm, confirming the presence of bioactive vesicles (Figure 2).

### 3.3. Effect of SHED and SHED-CM on Body Weight

Body weight was monitored daily for eight consecutive days. The cisplatin group exhibited a progressive decline in weight, while both the SHED and SHED-CM groups showed an attenuation of weight loss. Although weight reduction was observed in all groups, no statistically significant differences were detected between the SHED and SHED-CM groups (Figure 3).

### 3.4. Effect of SHED and SHED-CM on Kidney Weight

Kidney weight was significantly higher in the cisplatin group due to inflammation and edema compared to both treatment groups. The SHED-CM group exhibited a greater reduction in kidney weight compared to the SHED group, suggesting a stronger therapeutic effect (Figure 4). Statistical analysis showed that kidney weight was significantly lower in the SHED-CM group than in the cisplatin group (*p* < 0.05).

### 3.5. Macroscopic Kidney Damage Assessment

Gross examination of kidney samples revealed three distinct regions: necrotic areas (darkened portions), ischemic areas (paler than necrotic regions), and normal kidney tissue. The extent of damage was significantly lower in the SHED and SHED-CM groups compared to the cisplatin group, with the SHED-CM group displaying the least macroscopic damage. Quantitative analysis indicated a significantly reduced percentage of affected tissue in the SHED-CM group compared to the cisplatin group (*p* < 0.05) (Figure 5 and Figure 6A–C).

### 3.6. Effect of SHED and SHED-CM on Urine Volume and Protein Concentration

Urine volume was significantly reduced in the cisplatin group compared to the control group. Administration of SHED and SHED-CM resulted in a marked increase in urine volume, with the SHED-CM group demonstrating the highest recovery. Similarly, urinary protein concentration was significantly elevated in the cisplatin group, indicating impaired renal function. Both SHED and SHED-CM treatments reduced urinary protein levels, with the SHED-CM group exhibiting a more pronounced effect. Statistical analysis confirmed that urine volume was significantly higher and urinary protein concentration was significantly lower in the SHED-CM group compared to the cisplatin group (*p* < 0.01) (Figure 7 and Figure 8).

### 3.7. Histological Evaluation and Kidney Injury Score

Histological analysis (Figure 9) revealed extensive tubular necrosis, loss of the brush border, tubular dilatation, acute cell swelling, and glomerular swelling in the cisplatin group. The severity of these histopathological changes was significantly reduced in the SHED and SHED-CM groups, with the SHED-CM group displaying the least structural damage. Tubular injury scoring demonstrated a significantly lower injury score in the SHED-CM group compared to the cisplatin group (*p* < 0.05), indicating superior renal protection (Figure 10).

## 4. Discussion

AKI remains a major clinical challenge due to its high prevalence, morbidity, and mortality, with limited effective treatments available [7]. Cisplatin-induced nephrotoxicity is a well-established model for studying AKI, given its ability to mimic the tubular damage, oxidative stress, and inflammation observed in clinical AKI cases [6]. In this study, both SHED and SHED-CM demonstrated significant protective effects against cisplatin-induced renal damage, confirming previous evidence on the regenerative potential of MSC-based therapies [8,9,10]. Similar findings have been reported by Ashour et al., who demonstrated that both allogenic and xenogeneic MSCs significantly ameliorated cisplatin-induced AKI in rats, reducing oxidative stress markers and improving renal histology [11].

One of the critical findings of this study is that SHED-CM exerted similar or even superior renoprotective effects compared to SHED. This suggests that SHED mediate their therapeutic effects primarily through paracrine mechanisms, rather than direct cell integration into renal tissue. These paracrine effects are largely attributed to extracellular vesicles (EVs), including exosomes and microvesicles, which carry a range of bioactive molecules such as proteins, mRNAs, and microRNAs capable of modulating inflammation, apoptosis, and tissue regeneration [14]. In line with these results, Abouelkheir et al. reported that MSC-conditioned medium provided therapeutic effects comparable to MSC transplantation in cisplatin-induced AKI, with reduced tubular injury, apoptosis, and interstitial fibrosis [13]. Several studies have supported this mechanism, demonstrating that MSC-derived EVs and conditioned media are sufficient to reproduce many of the regenerative effects seen with whole-cell therapies [12,13,24,25].

Although the present study was not designed to dissect mechanism, converging evidence indicates that SHED secretome may confer renoprotection via immunomodulation, particularly by reprogramming macrophage responses. In I/R–AKI, SHED treatment attenuated renal infiltration of macrophages/neutrophils and reduced MCP-1, MIP-2 and IL-1β in vivo, while SHED-CM suppressed MCP-1 production in tubular epithelial cells, consistent with a paracrine anti-inflammatory effect in the kidney [23]. Beyond the kidney, SHED-CM and SHED-derived EVs have been shown to skew macrophages/microglia toward reparative M2 phenotypes and dampen NF-κB/MAPK signaling, thereby limiting tissue injury [29,30,31]. Recent work further identifies miRNA cargo as a driver of this effect: SHED-sEV miR-200c-3p promotes M2 polarization via PTEN/PI3K/Akt in vitro [32]. In AKI models, MSC-EVs can ameliorate injury by promoting M2 polarization and suppressing NF-κB signaling in macrophages, supporting an immunoregulatory route to renoprotection [33,34]. These reports provide a biologically plausible framework in which SHED-CM reduces early inflammatory injury and facilitates repair, potentially through macrophage reprogramming in the injured kidney.

Cisplatin-induced AKI is characterized by tubular epithelial cell death, inflammation, oxidative stress, and endothelial dysfunction [6]. The histopathological examination in this study revealed that cisplatin caused severe tubular necrosis, loss of brush borders, dilatation of tubules, and cellular swelling, consistent with previous reports [6,24]. However, both SHED and SHED-CM significantly ameliorated these histological alterations, with the SHED-CM group showing the lowest injury scores. These findings suggest that SHED-CM contains factors capable of preserving tubular architecture and mitigating cellular injury. Similar evidence was presented by Kim et al., [24] who demonstrated that MSC therapy reduced apoptosis and preserved kidney function in a cisplatin-induced AKI model, supporting the paracrine role of MSCs in renal repair. Notably, previous studies using MSCs and their EVs have shown that these products modulate the inflammatory microenvironment of injured kidneys by suppressing pro-inflammatory cytokines like TNF-α and IL-17, while enhancing anti-inflammatory cytokines such as IL-10 and IL-6 [12,13,24]. This dual action may explain the improved histological appearance and reduced injury scores observed in the SHED-CM group. Consistent with these observations, Simovic Markovic et al. showed that MSC administration in cisplatin nephrotoxicity models attenuated kidney injury through modulation of inflammatory cytokines, including reduction in TNF-α and IL-17, and enhancement of IL-10 and nitric oxide production [12].

The reduction in kidney weight in the SHED and SHED-CM groups compared to the cisplatin group suggests a decrease in renal edema and inflammation. Kidney swelling is a hallmark of cisplatin-induced injury due to cellular damage and inflammatory infiltration [6]. Therefore, the lower kidney weight observed in treated groups implies a therapeutic effect in reducing renal inflammation and restoring normal tissue architecture. Moreover, macroscopic examination revealed visibly less necrotic and ischemic areas in kidneys treated with SHED and SHED-CM, further supporting these observations.

In terms of functional parameters, urine output and proteinuria are critical indicators of renal function and glomerular integrity. Cisplatin-induced AKI resulted in a marked reduction in urine output and a significant increase in urinary protein concentration, reflecting tubular and glomerular dysfunction. Treatment with SHED and SHED-CM reversed these changes, as evidenced by increased urine volumes and reduced proteinuria. In addition, Yao et al. reported that human adipose-derived MSCs ameliorated cisplatin-induced AKI through anti-apoptotic pathways, which aligns with histopathological findings showing reduced tubular necrosis and preserved renal architecture following MSC-based treatments [25]. Notably, while both treatments were effective, SHED-CM showed a slightly better effect in restoring these functional markers, although the difference was not statistically significant. This is in line with previous studies where MSC-CM restored renal function by enhancing tubular cell proliferation, reducing oxidative stress, and decreasing apoptosis [13,24,25].

The therapeutic superiority of SHED-CM over SHED observed in this study is also consistent with studies indicating that conditioned medium may deliver therapeutic factors more effectively than whole-cell therapies [13,24]. One of the challenges associated with direct MSC transplantation includes low cell retention, poor survival in damaged tissues, risk of immune reactions, and potential for microvascular occlusion [15]. By contrast, cell-free therapies such as SHED-CM can bypass these limitations while retaining the therapeutic components, notably EVs, growth factors, and cytokines.

Importantly, SHED offer distinct advantages compared to other MSC sources, such as bone marrow or adipose tissue-derived stem cells. SHED can be collected non-invasively, have higher proliferative rates, and secrete greater amounts of bioactive molecules involved in regeneration [16,17,18]. Studies have demonstrated SHED’s effectiveness in models of bone defects, neural injuries, and ischemia, highlighting their versatility as a stem cell source [19,20,21,22,23]. The therapeutic potential of SHED has also been highlighted by Hattori et al. [23], who demonstrated that SHED could protect against AKI and promote renal regeneration through secretion of trophic factors and EVs. The present study builds on this knowledge by demonstrating that SHED-CM is not only effective in AKI but may also represent a more practical and safer alternative to cell-based therapies.

Another important consideration is the potential for standardizing SHED-CM production. Unlike live cell therapies, conditioned medium can be produced, stored, and transported more easily, facilitating its clinical use. Previous studies [13,15] have further emphasized that the use of conditioned medium or EVs avoids complications associated with direct stem cell transplantation, such as vascular occlusion or immune responses, supporting the role of SHED-CM as a safer alternative Moreover, the avoidance of cell transplantation reduces the risk of immune rejection and tumor formation, concerns that are particularly important in immunocompromised or severely ill patients [15].

Previous studies conducted by the same research team using Wistar rat models of renal injury have consistently reported that untreated or sham-operated animals exhibit preserved renal morphology, both macroscopically and microscopically [35,36]. These findings establish a well-documented baseline of normal renal architecture in healthy animals and provide a reliable reference for interpreting histological alterations in the present study. Consequently, the absence of a non-cisplatin control group is unlikely to compromise the validity of the tubular injury scoring or the statistical comparisons performed among the experimental groups. Taken together, the results of this study align with growing evidence that MSC-CM and EV-based therapies hold great promise in regenerative medicine. SHED-CM’s ability to reduce structural and functional kidney injury, as demonstrated here, supports its potential as a novel therapeutic agent for AKI. However, further studies are needed to characterize the specific components of SHED-CM responsible for these effects, including profiling of EV contents and understanding their mechanisms of action in renal tissue repair.

Several limitations should be acknowledged. First, key serum biochemical markers such as serum creatinine and blood urea nitrogen (BUN) were not measured, although urine volume, proteinuria, and histological injury scores provided relevant functional information. Inclusion of these biochemical parameters would have enhanced the clinical relevance of the findings. Second, the characterization of SHED-CM was relatively superficial. Scanning electron microscopy confirmed the presence of extracellular vesicles and protein concentration was quantified, but no quantitative profiling by nanoparticle tracking analysis (NTA) or dynamic light scattering (DLS) was performed, nor were canonical exosomal markers (CD9, CD63, CD81, TSG101, Alix) assessed. A comparative SEM image of unconditioned medium was also not obtained, limiting visualization of background particles. Third, although functional and histological improvements were demonstrated, the molecular mechanisms underlying these effects were not investigated, and specific extracellular vesicle–associated cargo or signaling pathways responsible for renoprotection remain undefined. Fourth, the experimental design did not include a healthy (non-cisplatin) control group or a vehicle-only injection group, which would have provided baseline values and excluded procedural effects. Fifth, only a single dose of SHED and SHED-CM was tested, which prevented determination of the optimal therapeutic window or assessment of dose-dependent toxicity. Sixth, the study endpoint at day 8 allowed evaluation of acute effects but not the long-term durability of renal protection, fibrosis prevention, or recurrence of injury. Finally, the translational potential of the findings is limited by differences between rodents and humans in immune responses, renal physiology, mesenchymal stem cell (MSC) behavior, and cisplatin metabolism. Future investigations should therefore include comprehensive biochemical profiling, standardized EV characterization following MISEV2018 guidelines, mechanistic analyses, dose-ranging studies, extended follow-up, healthy and vehicle-only control groups, and validation in large-animal models and human trials to strengthen reproducibility, clarify mechanisms, and establish clinical applicability.

## 5. Conclusions

In summary, this study demonstrates that SHED and SHED-CM provide significant protective effects in a rat model of cisplatin-induced AKI. Both treatments improved kidney function, reduced structural damage, and ameliorated histopathological alterations, with SHED-CM showing comparable or even superior effects to direct SHED injection. These findings suggest that the beneficial effects of SHED are largely mediated by paracrine mechanisms, supporting the use of SHED-CM as a cell-free, practical, and effective therapeutic option for AKI. Given its advantages in preparation, storage, and safety profile, SHED-CM may represent a promising candidate for clinical application in AKI and potentially other renal pathologies. Future research should focus on identifying the active components of SHED-CM and elucidating their mechanisms of action to optimize therapeutic strategies.

## Figures and Tables

**Figure 1 biology-14-01305-f001:**
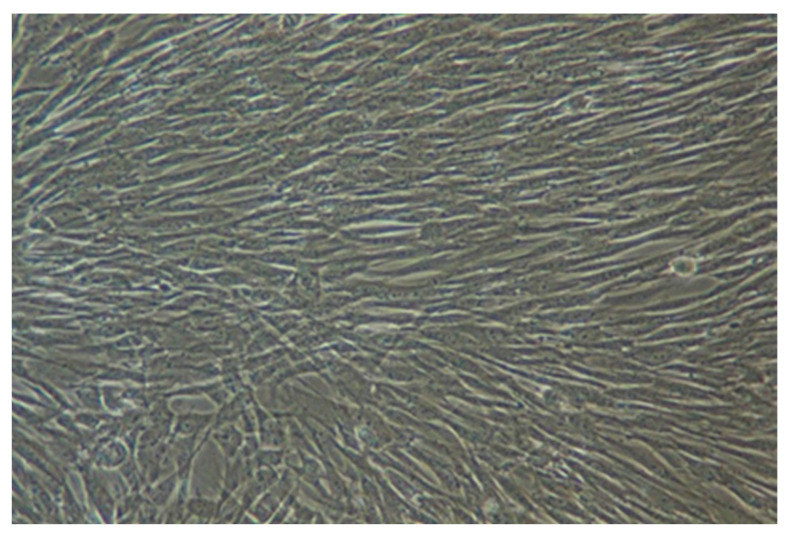
Stem cells from human exfoliated deciduous teeth (SHED) showing a homogeneous population of spindle-shaped, fibroblast-like cells under an inverted microscope.

**Figure 2 biology-14-01305-f002:**
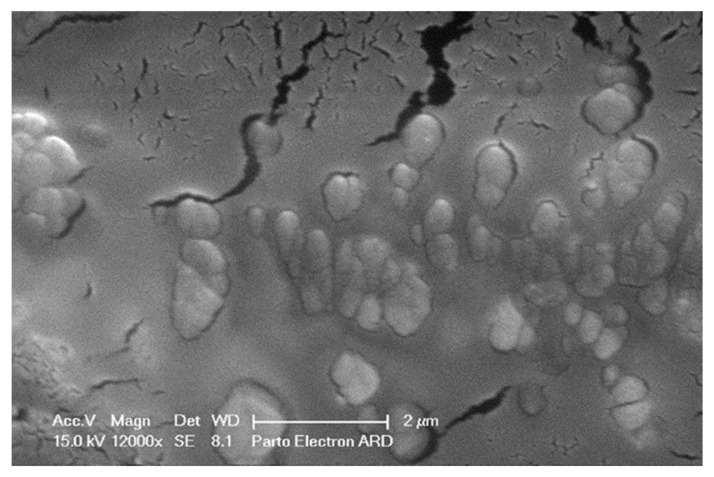
Scanning electron microscopy (SEM) image of extracellular vesicles present in SHED-conditioned medium (SHED-CM). Vesicles with sizes ranging from 200 nm to 1 μm are visible in the culture medium. Magnification: ×12,000.

**Figure 3 biology-14-01305-f003:**
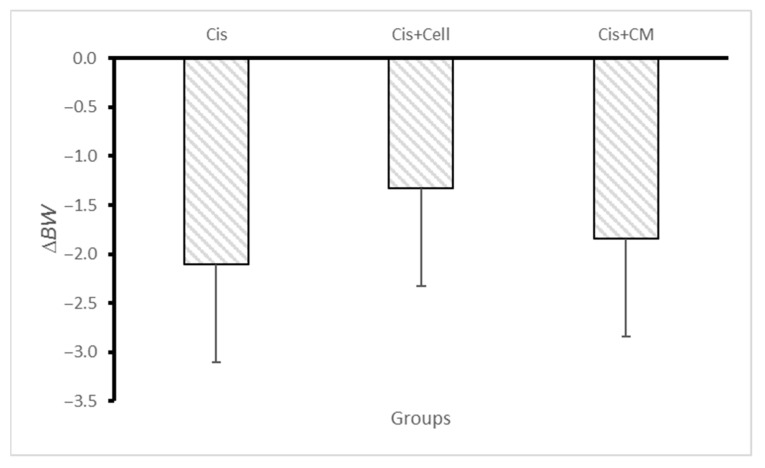
Mean weight loss in rats from the cisplatin, SHED, and SHED-conditioned medium (SHED-CM) groups over eight days. No significant difference in weight loss was observed among the groups. Data are expressed as mean ± SD; n = 7 rats/group; *p* < 0.05.

**Figure 4 biology-14-01305-f004:**
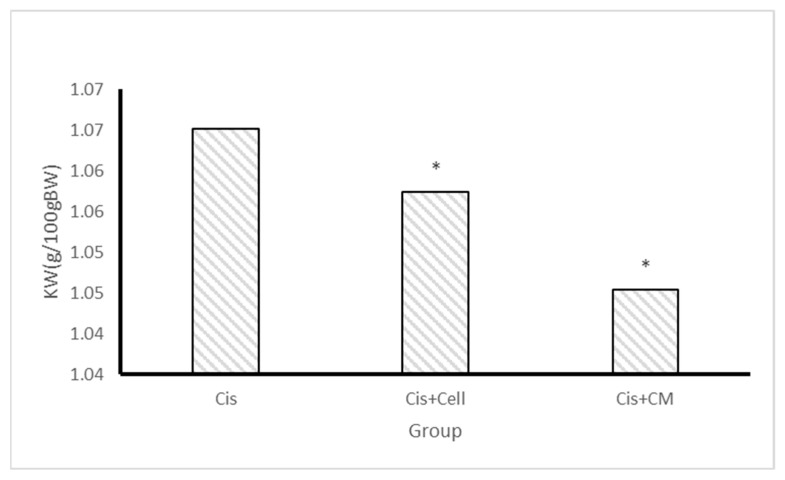
Kidney weight in the cisplatin, SHED, and SHED-conditioned medium (SHED-CM) groups. The kidney weight in both SHED and SHED-CM groups was lower than in the cisplatin group, with the SHED-CM group showing the greatest reduction. Data are expressed as mean ± SD; n = 7 rats/group. The asterisk (*) indicates a statistically significant difference compared with the cisplatin group (*p* < 0.05).

**Figure 5 biology-14-01305-f005:**
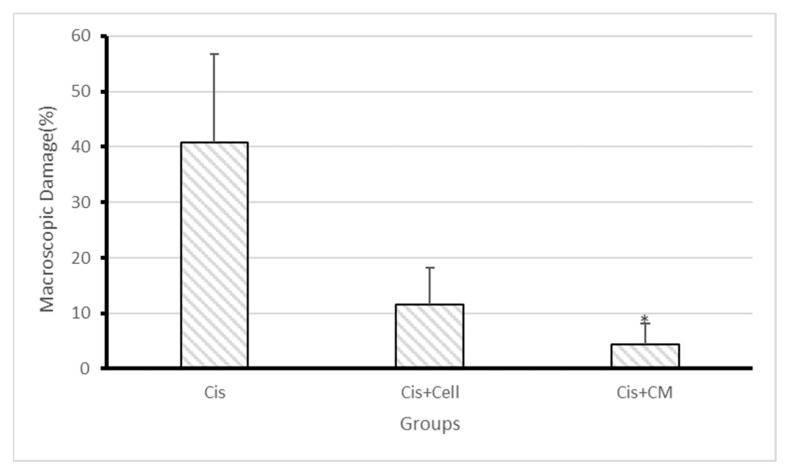
Percentage of macroscopic kidney damage (necrosis and ischemia) in the cisplatin, SHED, and SHED-conditioned medium (SHED-CM) groups. The extent of macroscopic damage was significantly lower in the SHED and SHED-CM groups compared to the cisplatin group, with the SHED-CM group showing the least damage. Data are expressed as mean ± SD; n = 7 rats/group. The asterisk (*) indicates a statistically significant difference compared with the cisplatin group (*p* < 0.05).

**Figure 6 biology-14-01305-f006:**
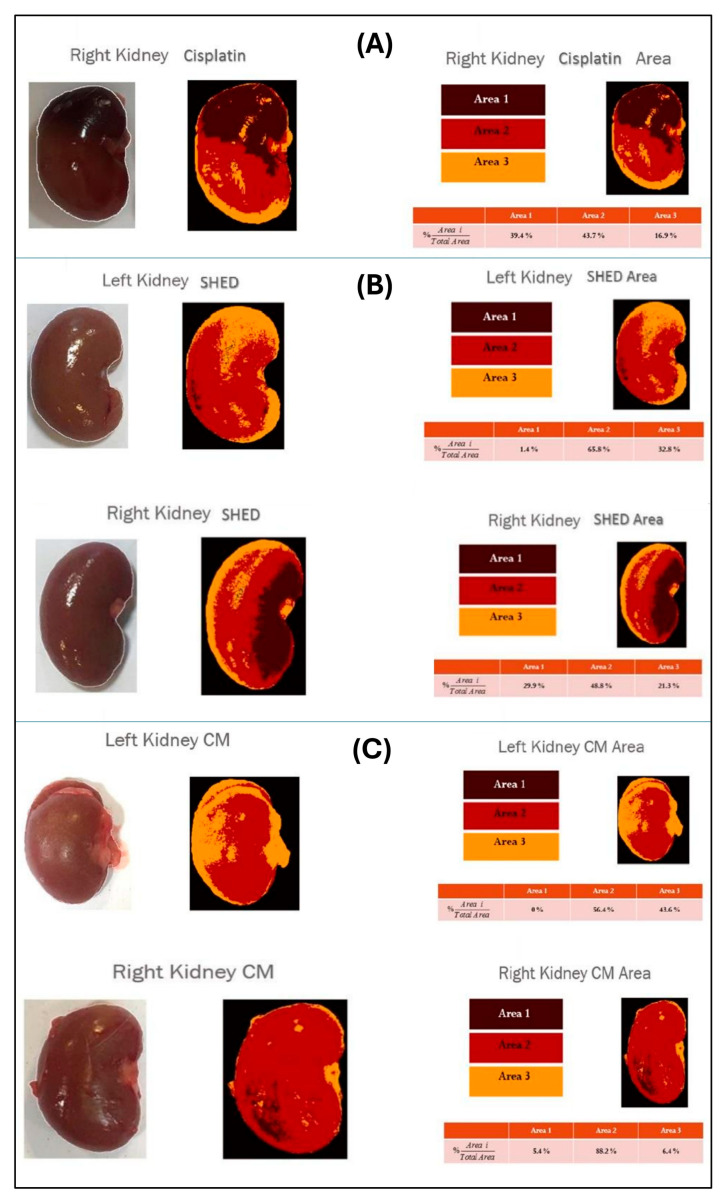
Representative macroscopic images of kidney damage in the three experimental groups. Area 1 represents necrotic tissue, Area 2 represents ischemic tissue, and Area 3 represents normal kidney tissue. (**A**) Cisplatin group showing extensive necrotic and ischemic areas. (**B**) SHED group showing reduced necrotic and ischemic regions compared with the cisplatin group. (**C**) SHED-conditioned medium (SHED-CM) group showing the least macroscopic kidney damage. Analysis was performed on all rats, and one representative image from each group is shown. In the cisplatin group, the right kidney was selected for macroscopic visualization due to clearer necrotic demarcation, while histological analyses were performed on both kidneys.

**Figure 7 biology-14-01305-f007:**
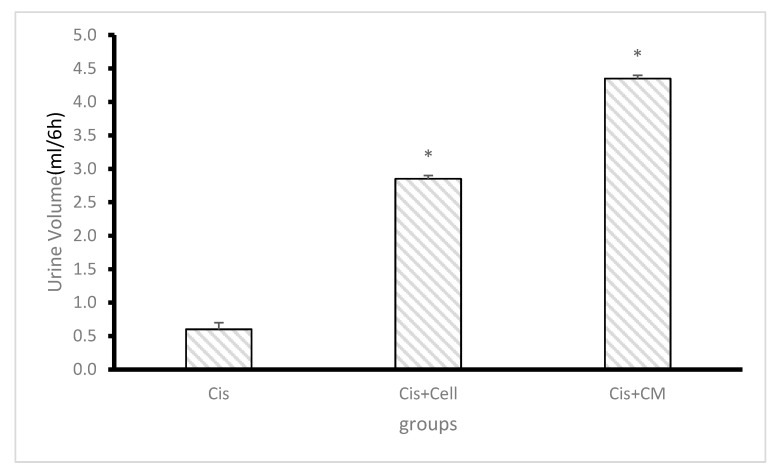
Effect of cisplatin, SHED, and SHED-conditioned medium (SHED-CM) on urine volume. Cisplatin administration significantly reduced urine volume compared to treated groups. No significant difference was observed between the SHED and SHED-CM groups. Data are expressed as mean ± SD; n = 7 rats/group. The asterisk (*) indicates a statistically significant difference compared with the cisplatin group (*p* < 0.05).

**Figure 8 biology-14-01305-f008:**
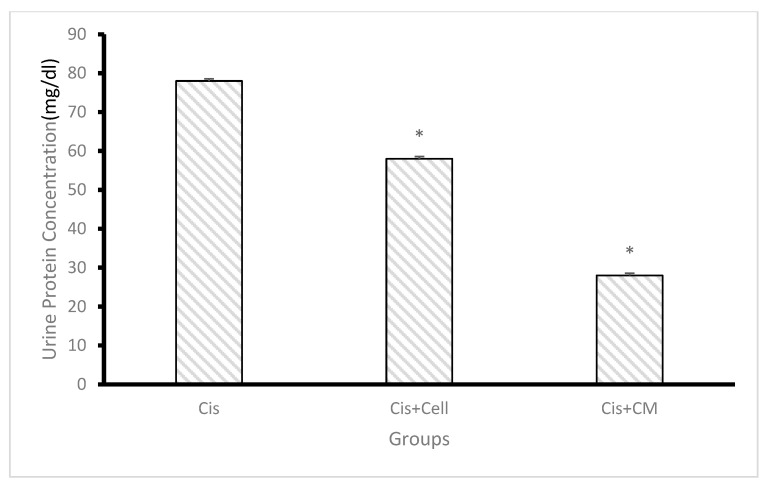
Effect of cisplatin, SHED, and SHED-conditioned medium (SHED-CM) on urinary protein concentration. Cisplatin administration significantly increased urinary protein concentration compared to the treated groups. No significant difference was observed between the SHED and SHED-CM groups. Data are expressed as mean ± SD; n = 7 rats/group. The asterisk (*) indicates a statistically significant difference compared with the cisplatin group (*p* < 0.05).

**Figure 9 biology-14-01305-f009:**
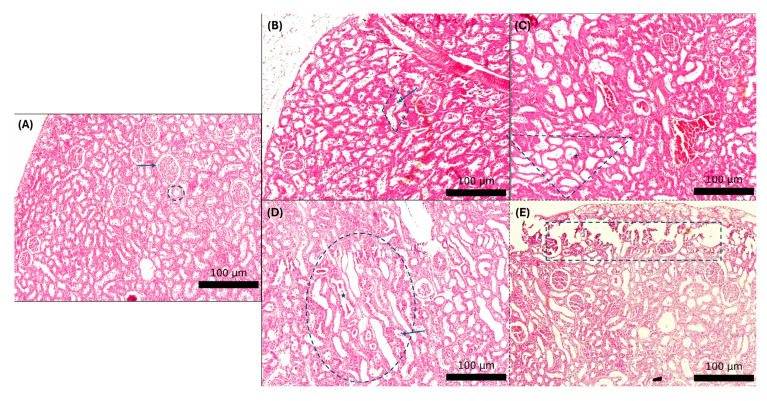
Representative H&E-stained kidney sections illustrating the tubular injury grading reference used in this study. (**A**) Grade 0 (control): preserved tubular architecture with intact brush borders and no casts or interstitial inflammation. Arrow shows glomerulus and circle shows normal tubule. (**B**) Grade 1 (mild): Rhombic shows focal tubular epithelial flattening with early loss of brush border and minimal interstitial change and mild tubular dilatation. Arrow shows luminal debris. (**C**) Grade 2 (moderate): Triangle shows patchy proximal tubular necrosis with tubular dilatation and star shows occasional intraluminal casts. (**D**) Grade 3 (marked): Oval shows interstitial edema and widespread tubular necrosis and dilatation. Star shows prominent granular/hyaline casts, and arrow shows luminal debris. (**E**) Grade 4 (severe): Rectangular shows extensive cortical tubular necrosis and dilatation with epithelial denudation.

**Figure 10 biology-14-01305-f010:**
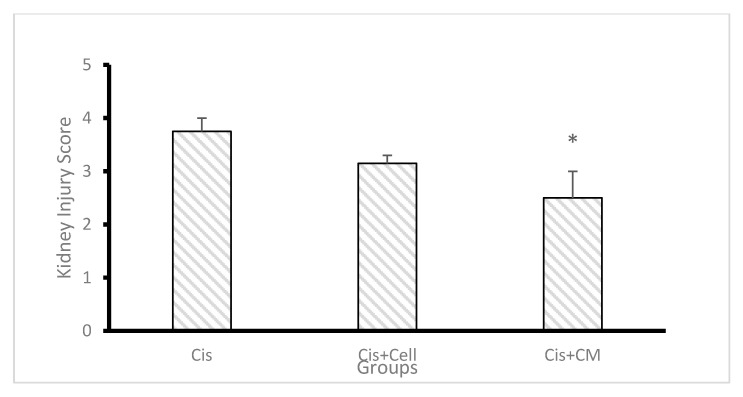
Effect of SHED, SHED-conditioned medium (SHED-CM), and cisplatin on kidney injury score. The mean injury score was highest in the cisplatin group and significantly lower in the SHED-CM group, indicating the greatest protective effect. Data are expressed as mean ± SD; n = 7 rats/group. The asterisk (*) indicates a statistically significant difference compared with the cisplatin group (*p* < 0.05).

## Data Availability

The data presented in this study are available on request from the corresponding author.

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
