# Peer review of "Therapeutic Effects of Stem Cells from Human Exfoliated Deciduous Teeth and Their Conditioned Medium in Cisplatin-Induced Acute Kidney Injury: An In Vivo Study"

_biology, 2025, doi:10.3390/biology14091305_

Round 1

Reviewer 1 Report

Comments and Suggestions for Authors

The manuscript entitled "Therapeutic effects of stem cells from human exfoliated deciduous teeth and their conditioned medium in cisplatin-induced acute kidney injury: An in vivo study" by Talebi et al. report the beneficial and regenerative effects of EVs from human exfoliated deciduous teeth on acute kidney injury. The topic is interesting, but in order to sustain their hypothesis and conclusions, the Authors should address this important point:

-The histological evaluation reported in Fig. 9 needs to be accompanied with representative images of hematoxylin and eosin (H&E) stained-tissue sections for each experimental group.

Author Response

REVIEWER 1:

The manuscript entitled "Therapeutic effects of stem cells from human exfoliated deciduous teeth and their conditioned medium in cisplatin-induced acute kidney injury: An in vivo study" by Talebi et al. report the beneficial and regenerative effects of EVs from human exfoliated deciduous teeth on acute kidney injury. The topic is interesting, but in order to sustain their hypothesis and conclusions, the Authors should address this important point:

-The histological evaluation reported in Fig. 9 needs to be accompanied with representative images of hematoxylin and eosin (H&E) stained-tissue sections for each experimental group.

RESPONSE: We sincerely appreciate your valuable feedback regarding the histological evaluation presented. The initial decision not to include representative histological images for each experimental group was intentional, as the study was designed to prioritize a quantitative approach over purely qualitative visual interpretation. A tubular injury scoring system was applied to convert qualitative observations into objective numerical data, allowing for clearer comparisons between groups and making the results more accessible and easier to interpret for a broader readership, including those with limited experience in histopathology. This approach was intended to avoid overwhelming readers with multiple histological images and instead provide a concise, evidence-based summary through quantitative scoring. However, following your constructive suggestion, the revised manuscript now includes representative H&E-stained images in a new Figure. These images are not intended to depict findings for each experimental group individually, but rather to illustrate the grading criteria used during the blinded histological assessment. By presenting examples of the morphological features corresponding to each injury grade, readers can better understand how the scoring system was applied while maintaining the focus on the quantitative evaluation as the primary analytical method. Importantly, the Grade 0 images used as a reference for normal renal morphology were obtained from previous investigations conducted by the same research team under identical histological processing and imaging conditions, ensuring methodological consistency and validity.

Reviewer 2 Report

Comments and Suggestions for Authors

This manuscript investigates the therapeutic efficacy of SHED and SHED-derived conditioned medium (SHED-CM) in a cisplatin-induced acute kidney injury (AKI) rat model. The topic is of interest and the approach is innovative, particularly in exploring the potential of a cell-free therapy. However, the study suffers from multiple critical limitations that compromise its scientific rigor and mechanistic depth. These include insufficient mechanistic validation, limited biochemical assessments, inadequate characterization of the therapeutic product, short observational duration, and a lack of transparency regarding cell source variability. These issues must be addressed to enhance the validity and reproducibility of the findings.

  1. The manuscript does not explore how SHED-CM mediates its renoprotective effects. In contrast to prior studies emphasizing immunomodulatory actions of SHED via macrophage polarization, this work lacks any assessment of immune cell involvement or anti-inflammatory pathways.
  2. The functional evaluation of renal injury is restricted to urine volume, proteinuria, and histological injury scores. These endpoints, while relevant, are insufficient to comprehensively assess renal recovery. Crucial biochemical markers such as serum creatinine and BUN are not reported, limiting clinical relevance.
  3. The characterization of SHED-CM is superficial. While SEM imaging indicates the presence of extracellular vesicles, no quantitative data are provided on particle concentration, size distribution, or the expression of standard exosomal markers. The bioactive content of the CM remains undefined, and the lack of standardized profiling undermines reproducibility and mechanistic interpretation.
  4. The study endpoint at day 8 is insufficient to evaluate the durability of therapeutic effects. Without longer-term observation, it is unclear whether the treatment provides lasting renal protection or merely delays injury. 
  5. The source of SHED is poorly documented. The manuscript does not indicate whether the cells were obtained from a single donor or pooled from multiple individuals, nor does it describe donor demographics. This omission limits reproducibility and prevents assessment of biological variability, both of which are critical for translational applications.

Author Response

This manuscript investigates the therapeutic efficacy of SHED and SHED-derived conditioned medium (SHED-CM) in a cisplatin-induced acute kidney injury (AKI) rat model. The topic is of interest, and the approach is innovative, particularly in exploring the potential of a cell-free therapy. However, the study suffers from multiple critical limitations that compromise its scientific rigor and mechanistic depth. These include insufficient mechanistic validation, limited biochemical assessments, inadequate characterization of the therapeutic product, short observational duration, and a lack of transparency regarding cell source variability. These issues must be addressed to enhance the validity and reproducibility of the findings.

  1. The manuscript does not explore how SHED-CM mediates its renoprotective effects. In contrast to prior studies emphasizing immunomodulatory actions of SHED via macrophage polarization, this work lacks any assessment of immune cell involvement or anti-inflammatory pathways.

RESPONSE: Thank you for your insightful comment. We agree that the current manuscript did not directly investigate the involvement of immune cell modulation or macrophage polarization. Our primary objective was to evaluate the therapeutic efficacy of SHED-CM in cisplatin-induced AKI rather than elucidating the underlying molecular mechanisms. However, we have now revised the Discussion section to include an explanation of the potential immunomodulatory pathways based on evidence from previous studies, emphasizing the role of SHED-CM in modulating inflammation and macrophage polarization.

  1. The functional evaluation of renal injury is restricted to urine volume, proteinuria, and histological injury scores. These endpoints, while relevant, are insufficient to comprehensively assess renal recovery. Crucial biochemical markers such as serum creatinine and BUN are not reported, limiting clinical relevance.

RESPONSE: Thank you for your valuable comment. We acknowledge that our functional evaluation of renal injury was limited to urine volume, proteinuria, and histological injury scores. Measurement of additional biochemical markers, such as serum creatinine and blood urea nitrogen (BUN), would indeed provide a more comprehensive assessment of renal recovery and enhance the clinical relevance of the findings. However, due to limitations in available resources and the study design, we did not perform these assays. We have now revised the Discussion section to explicitly acknowledge this limitation and highlight the need for future studies to incorporate these biochemical parameters.

  1. The characterization of SHED-CM is superficial. While SEM imaging indicates the presence of extracellular vesicles, no quantitative data are provided on particle concentration, size distribution, or the expression of standard exosomal markers. The bioactive content of the CM remains undefined, and the lack of standardized profiling undermines reproducibility and mechanistic interpretation.

RESPONSE: Thank you for this constructive comment. We agree that a more detailed characterization of SHED-CM, including quantitative extracellular vesicle (EV) profiling and bioactive content analysis, would strengthen mechanistic interpretation and improve reproducibility. In this study, we confirmed the presence of extracellular vesicles in SHED-CM using scanning electron microscopy (SEM) and quantified total protein concentration via BCA assay, which provided preliminary evidence of EV presence and bioactive content. However, due to resource limitations, we did not perform nanoparticle tracking analysis (NTA), dynamic light scattering (DLS), or Western blotting for exosomal markers (e.g., CD9, CD63, CD81, TSG101, Alix). We have now revised the Discussion section to acknowledge this limitation and propose including standardized EV profiling and proteomic analyses in future studies.

  1. The study endpoint at day 8 is insufficient to evaluate the durability of therapeutic effects. Without longer-term observation, it is unclear whether the treatment provides lasting renal protection or merely delays injury. 

RESPONSE: Thank you for highlighting this important point. We agree that the short follow-up period represents a limitation of the present study. Our primary objective was to evaluate the early therapeutic effects of SHED and SHED-CM in the acute phase of cisplatin-induced AKI, which typically peaks between days 5 and 8 in rodent models. Therefore, we selected day 8 as the endpoint to capture the peak injury and early recovery phase. Although the duration was limited, the presented data—including kidney weight, kidney-to-body weight ratio, urine volume, proteinuria, and macroscopic histological injury scores—collectively allowed us to meet the core objectives of the study and demonstrate early therapeutic efficacy. It is important to note that this project was conducted as part of a graduate thesis under strict budgetary constraints, which limited the possibility of extended follow-up. However, we acknowledge that this design does not allow us to determine the long-term durability of renal protection or potential prevention of fibrosis. We have now addressed this limitation in the Discussion and suggested that future studies incorporate extended follow-up to evaluate sustained outcomes.

  1. The source of SHED is poorly documented. The manuscript does not indicate whether the cells were obtained from a single donor or pooled from multiple individuals, nor does it describe donor demographics. This omission limits reproducibility and prevents assessment of biological variability, both of which are critical for translational applications.

RESPONSE: Thank you for raising this point. The SHED used in this study were obtained from the Iranian Banking and Developing SHED Knowledge-Based Company (Dental Stem Cell Bank). According to the supplier's documentation, the vials used were pooled SHED samples derived from three healthy donors, each aged between 6 and 8 years, collected under standardized protocols with parental consent. However, individual donor demographic information beyond age range and health status was not provided to us due to anonymization policies. We have now revised the Materials and Methods section to clarify the source and donor origin of the SHED used in this study.

Reviewer 3 Report

Comments and Suggestions for Authors

The manuscript entitled “Therapeutic effects of stem cells from human exfoliated deciduous teeth and their conditioned medium in cisplatin-induced 3 acute kidney injury: An in vivo study” is reviewed. The following comments may be considered:

  1. In Fig. 2, SEM images of extracellular vesicles present in SHED-conditioned medium (SHED-CM) are shown. Can authors show a comparative control image without conditioned medium?
  2. In section 2.3, SHED Isolation and Culture, the author mentioned that SHED cells were obtained from the Iranian Banking and Developing SHED 136 Knowledge-Based Company (Dental Stem Cell Bank). Isolation and characterisation details are missing. Can the author mention the isolation procedure and its characterisation in this section?
  3. In the study design, the authors have taken 21 male Wistar rats and randomised them into three groups. (i) Cisplatin, (ii) Cis+SHED (500,000 cells/1 mL per injection), and (iii) Cis+SHED-conditioned medium (1 mL per injection). In this study design, the normal control group is missing. Without a normal control group, the application of statistics to the disease group to compare treatment groups is difficult. Also, histological scoring is difficult to apply without a normal control group.
  4. In Fig.6, why has the author not taken the Left Kidney cisplatin for necrotic area visualisation?
  5. The data is not sufficient to demonstrate the efficacy. No histopathology, no mechanistic findings, and experimental design without a normal control for comparison purposes.

Author Response

The manuscript entitled “Therapeutic effects of stem cells from human exfoliated deciduous teeth and their conditioned medium in cisplatin-induced 3 acute kidney injury: An in vivo study” is reviewed. The following comments may be considered:

  1. In Fig. 2, SEM images of extracellular vesicles present in SHED-conditioned medium (SHED-CM) are shown. Can authors show a comparative control image without conditioned medium?

RESPONSE: Thank you for your thoughtful observation. In this study, our primary objective was to confirm the presence and morphology of extracellular vesicles in SHED-conditioned medium. For this reason, we did not capture SEM images of unconditioned DMEM or PBS as negative controls. We acknowledge that presenting a comparative control image would improve clarity by distinguishing vesicle-specific structures from potential background artifacts. However, because we employed a specialized exosome isolation kit and considering the well-established absence of EVs in PBS or unconditioned DMEM, we considered additional SEM imaging of control media to be unnecessary for the morphological confirmation performed here. Moreover, this study was conducted as part of a graduate thesis under budgetary constraints, which limited our ability to expand imaging resources. Nonetheless, we recognize this omission as a limitation and have revised the Discussion section accordingly. We also propose that future studies incorporate comparative SEM imaging of both conditioned and unconditioned media to enhance clarity, reproducibility, and interpretation.

  1. In section 2.3, SHED Isolation and Culture, the author mentioned that SHED cells were obtained from the Iranian Banking and Developing SHED 136 Knowledge-Based Company (Dental Stem Cell Bank). Isolation and characterisation details are missing. Can the author mention the isolation procedure and its characterisation in this section?

RESPONSE: Thank you for this valuable comment. The SHED used in our study were obtained from the Iranian Banking and Developing SHED Knowledge-Based Company (Dental Stem Cell Bank), where they were isolated, expanded, and characterized before delivery. We have now updated Section 2.3 to include a concise description of the isolation protocol and characterization process based on the supplier’s standardized procedures, which are in accordance with previously published methods.

  1. In the study design, the authors have taken 21 male Wistar rats and randomised them into three groups. (i) Cisplatin, (ii) Cis+SHED (500,000 cells/1 mL per injection), and (iii) Cis+SHED-conditioned medium (1 mL per injection). In this study design, the normal control group is missing. Without a normal control group, the application of statistics to the disease group to compare treatment groups is difficult. Also, histological scoring is difficult to apply without a normal control group.

RESPONSE: We appreciate your thoughtful comment and fully agree that including a healthy (non-cisplatin) control group would have provided valuable baseline reference data for statistical comparisons and histological scoring. The primary objective of this first-phase study was to evaluate the therapeutic effects of SHED and SHED-CM in a cisplatin-induced AKI model. To minimize animal use in accordance with ethical guidelines and due to significant budgetary constraints—particularly the high cost associated with exosome isolation and characterization—the experimental groups were limited to those directly addressing the central research question (cisplatin only, cisplatin + SHED, and cisplatin + SHED-CM). Importantly, previous investigations conducted by the same research team using Wistar rat models of renal injury have consistently demonstrated that untreated animals exhibit normal renal morphology both macroscopically and microscopically. These prior findings, now cited in the revised manuscript, provided strong confidence that the absence of a normal control group would not compromise the interpretation of injury severity or therapeutic response in the current study. Furthermore, it was considered unjustifiable to sacrifice additional animals for the inclusion of a negative control group when comparable histological and functional information was already available from previous studies conducted under identical methodological conditions. Nonetheless, the absence of a healthy control group is acknowledged as a limitation, and the Discussion section has been revised accordingly, emphasizing that future studies will incorporate a non-cisplatin control group to provide more robust baseline comparisons.

  1. In Fig.6, why has the author not taken the Left Kidney cisplatin for necrotic area visualisation?

RESPONSE: Thank you for the observation regarding Figure 6. The images included in this figure were selected to effectively illustrate the extent of renal parenchymal damage and therapeutic responses across the experimental groups. In the cisplatin group, the right kidney was specifically chosen for macroscopic visualization because it consistently showed more distinct and reproducible areas of necrosis compared to the left kidney, thereby providing a clearer demonstration of tissue damage. In the SHED and SHED-CM groups, representative kidneys with the most characteristic macroscopic findings were selected for consistency. It should also be noted that the histological evaluation was performed on both kidneys for all groups, and representative H&E-stained sections have now been included in the revised manuscript to complement the macroscopic findings. These additions enhance the visualization of renal injury patterns without affecting the results or conclusions.

  1. The data is not sufficient to demonstrate the efficacy. No histopathology, no mechanistic findings, and experimental design without a normal control for comparison purposes.

RESPONSE: Thank you for this constructive criticism. We respectfully note that the present study included a comprehensive histopathological evaluation and tubular injury scoring, which revealed significant differences between groups. In addition to these analyses, several functional parameters—including kidney weight, kidney-to-body weight ratio, urine volume, and proteinuria—were assessed, collectively supporting the therapeutic efficacy of SHED and SHED-CM in reducing renal injury during the acute phase of cisplatin-induced AKI. The histological analysis was performed comprehensively, and representative H&E-stained images have now been added to the revised manuscript to enhance clarity. Furthermore, the histological assessment was converted from a purely qualitative interpretation of slides into a quantitative tubular injury grading system, making the results more objective and facilitating interpretation, particularly for readers with limited experience in evaluating H&E-stained renal sections. However, it is acknowledged that the lack of molecular mechanistic analyses (e.g., cytokine profiling, EV cargo characterization) and the absence of a normal healthy control group reduce the depth of interpretation and limit translational applicability. The inclusion of a healthy control group was not pursued in this first-phase study to minimize animal use in accordance with ethical guidelines and because extensive data from previous investigations conducted by the same research team have consistently demonstrated that untreated Wistar rats exhibit preserved renal morphology and normal functional parameters under identical experimental conditions. Given the availability of these validated reference data, sacrificing additional animals solely for morphological baseline confirmation was considered scientifically unnecessary and ethically unjustifiable. These limitations have been explicitly highlighted in the revised Discussion section, and future studies are planned to incorporate both detailed mechanistic assays and a non-cisplatin control group to strengthen baseline comparisons and enhance translational relevance.

Reviewer 4 Report

Comments and Suggestions for Authors

I carefully read and reviewed the manuscript titled "Therapeutic effects of stem cells from human exfoliated deciduous teeth and their conditioned medium in cisplatin-induced acute kidney injury: An in vivo study".  Authors studied the therapeutic potential of stem cells from human exfoliated deciduous teeth (SHED) and their conditioned medium (SHED-CM), a relatively novel and ethically accessible source of mesenchymal stem cells (MSCs), in acute kiney injury.  

The use of cisplatin-induced AKI in Wistar rats is a validated and widely used model to mimic nephrotoxicity-related kidney injury. Authors evaluated both functional (urine volume, proteinuria) and morphological (histopathological scoring) parameters in the study. 

Authors found that SHED-CM performed better than SHED injections. Thus, the therapeutic potential of cell-free therapy, which is advantageous in terms of safety, storage, and immunogenicity is highlighted by the study findings.

However, I noted several issues that must be revised:

- The study results were promising but findings in rodent models may not directly translate to human AKI due to differences in immune response, kidney physiology, and MSC behavior. Discuss please.

- A placebo control group receiving just the injection medium could help rule out any confounding effects due to the injection process itself. Explain and discuss please.

- Authors used a single dose of SHED and SHED-CM. Dose-ranging studies are needed to define optimal therapeutic windows and minimize toxicity. Emphasize on please.

- The authors assessed only acute outcomes. Long-term renal function, fibrosis, and recurrence of injury were not assessed, limiting conclusions about durability and safety of SHED-CM therapy. Discuss please.

- Authors showed functional and histological improvements in the study but they didn't emphasize the molecular mechanisms or identify specific extracellular vesicle (EV)-associated factors responsible for the observed effects. Discuss appropriately please.

The main question in the paper was whether stem cells from human exfoliated deciduous teeth (SHED) and their conditioned medium (SHED-CM) had any effects on cisplatin induced acute kidney injury model. The topic is novel and relevant to the field. It originally fills a gap in the literature. Because the study revealed that SHED-CM performed better than SHED injections and highlighted the therapeutic potential of cell-free therapy in AKI, which is advantageous in terms of safety, storage, and immunogenicity.

To improve methodology, authors should consider a control group receiving just the injection medium. This could help rule out any confounding effects due to the injection process itself.

Conclusions are justified.

Author Response

I carefully read and reviewed the manuscript titled "Therapeutic effects of stem cells from human exfoliated deciduous teeth and their conditioned medium in cisplatin-induced acute kidney injury: An in vivo study".  Authors studied the therapeutic potential of stem cells from human exfoliated deciduous teeth (SHED) and their conditioned medium (SHED-CM), a relatively novel and ethically accessible source of mesenchymal stem cells (MSCs), in acute kiney injury.  

The use of cisplatin-induced AKI in Wistar rats is a validated and widely used model to mimic nephrotoxicity-related kidney injury. Authors evaluated both functional (urine volume, proteinuria) and morphological (histopathological scoring) parameters in the study. 

Authors found that SHED-CM performed better than SHED injections. Thus, the therapeutic potential of cell-free therapy, which is advantageous in terms of safety, storage, and immunogenicity is highlighted by the study findings.

However, I noted several issues that must be revised:

- The study results were promising but findings in rodent models may not directly translate to human AKI due to differences in immune response, kidney physiology, and MSC behavior. Discuss please.

RESPONSE: Thank you for this important comment. We agree that findings from rodent models may not directly translate to human AKI due to differences in immune response, kidney physiology, and MSC behavior. Nonetheless, the cisplatin-induced AKI model in Wistar rats remains a widely accepted and validated platform for evaluating nephroprotective interventions. As in other animal studies exploring MSC-based therapies, our approach was designed as a foundational step to assess the safety and therapeutic efficacy of SHED and SHED-CM before advancing to larger animal models and, ultimately, clinical trials. This study represents the first phase of a multi-stage research program. Given ethical and financial constraints, it was essential to begin with a small-animal model to establish preliminary efficacy and optimize delivery protocols. Future phases will incorporate more translationally relevant models and mechanistic assays to bridge the gap between preclinical findings and human application. We have now revised the Discussion section to address this limitation and to clarify the sequential nature of our research plan.

- A placebo control group receiving just the injection medium could help rule out any confounding effects due to the injection process itself. Explain and discuss please.

RESPONSE: Thank you for this valuable observation. Including a placebo control group receiving only the injection medium would indeed have provided an additional safeguard to rule out potential confounding effects related to the injection procedure itself, such as local tissue trauma or inflammatory responses unrelated to SHED or SHED-CM administration. In the current study, the experimental design was focused on evaluating the therapeutic efficacy of SHED and SHED-CM in cisplatin-induced AKI, and to comply with ethical guidelines emphasizing the reduction of animal use, an additional placebo group was not included. The likelihood of significant confounding from the injection procedure is considered low because the same vehicle and injection route were consistently used across both treatment groups, minimizing variability attributable to the medium. Moreover, previous studies employing similar vehicles and routes have reported no histological or functional alterations caused solely by the injection medium, and the significant differences observed between SHED and SHED-CM groups further support that the therapeutic outcomes were attributable to the administered interventions rather than the injection process itself. Nevertheless, the absence of a placebo control group is recognized as a limitation, and this has been addressed in the revised Discussion section.

- Authors used a single dose of SHED and SHED-CM. Dose-ranging studies are needed to define optimal therapeutic windows and minimize toxicity. Emphasize on please.

RESPONSE: Thank you for this valuable comment. We agree that determining the optimal dosing regimen is critical to maximize therapeutic efficacy and minimize potential toxicity. In this study, we used a single dose of SHED and SHED-CM based on previously published protocols and pilot data, aiming to evaluate the initial therapeutic potential. As this work represents the first phase of a broader translational research plan, our primary goal was to establish proof-of-concept efficacy and safety in a small-animal model. We have now added a statement to the Discussion emphasizing the importance of conducting dose-ranging studies in future phases to identify optimal therapeutic windows and safety margins.

- The authors assessed only acute outcomes. Long-term renal function, fibrosis, and recurrence of injury were not assessed, limiting conclusions about durability and safety of SHED-CM therapy. Discuss please.

RESPONSE: Thank you for this important comment. We agree that our study primarily focused on acute outcomes to evaluate the early therapeutic effects of SHED and SHED-CM in cisplatin-induced AKI. This approach was intentional since cisplatin nephrotoxicity typically peaks between days 5 and 8 in rodent models, and our objective was to investigate initial safety and efficacy. However, we acknowledge that the absence of long-term follow-up limits our ability to draw conclusions regarding the durability, fibrosis prevention, and long-term safety of SHED-CM therapy. We have revised the Discussion section to highlight this limitation and propose extended longitudinal studies to evaluate sustained therapeutic effects and potential delayed complications.

- Authors showed functional and histological improvements in the study but they didn't emphasize the molecular mechanisms or identify specific extracellular vesicle (EV)-associated factors responsible for the observed effects. Discuss appropriately please.

RESPONSE: Thank you for your insightful comment. We agree that the current manuscript did not directly investigate the involvement of immune cell modulation or macrophage polarization. Our primary objective was to evaluate the therapeutic efficacy of SHED-CM in cisplatin-induced AKI rather than elucidating the underlying molecular mechanisms. Given the scope and resource limitations of this initial investigation, we prioritized outcome-based assessments over mechanistic profiling. However, we have now revised the Discussion section to include an explanation of the potential immunomodulatory pathways based on evidence from previous studies, emphasizing the role of SHED-CM in modulating inflammation and macrophage polarization. These next steps will be essential to elucidate the specific pathways through which SHED-CM exerts its therapeutic effects and to support its development as a cell-free regenerative therapy.

The main question in the paper was whether stem cells from human exfoliated deciduous teeth (SHED) and their conditioned medium (SHED-CM) had any effects on cisplatin induced acute kidney injury model. The topic is novel and relevant to the field. It originally fills a gap in the literature. Because the study revealed that SHED-CM performed better than SHED injections and highlighted the therapeutic potential of cell-free therapy in AKI, which is advantageous in terms of safety, storage, and immunogenicity. To improve methodology, authors should consider a control group receiving just the injection medium. This could help rule out any confounding effects due to the injection process itself. Conclusions are justified.

RESPONSE: We thank the reviewer for recognizing the novelty and relevance of this work, as well as for acknowledging that the conclusions are justified. The suggestion regarding the inclusion of a vehicle-only injection control group is well taken, as such a group would indeed help exclude potential confounding effects of the injection procedure itself. In the current study, the experimental design was limited to three groups to reduce animal use in accordance with ethical guidelines while directly addressing the central research question of whether SHED or SHED-CM exert renoprotective effects in a cisplatin-induced AKI model. Additionally, extensive data from previous investigations conducted by the same research team have consistently demonstrated that untreated Wistar rats exhibit preserved renal morphology and normal functional parameters under identical experimental conditions. Therefore, sacrificing additional animals for a medium-only control group was considered scientifically unnecessary and ethically unjustifiable. The absence of such a control group is acknowledged as a limitation, and the Discussion has been revised accordingly. Future studies are planned to incorporate vehicle-only injection groups alongside expanded mechanistic analyses to strengthen the experimental design and further validate the therapeutic potential of SHED and SHED-CM.

Round 2

Reviewer 1 Report

Comments and Suggestions for Authors

Self-citation references must be revised. No further comments.

Author Response

Reviewer #1:

Self-citation references must be revised. No further comments.

RESPONSE: We sincerely thank the reviewer for this valuable comment. We have carefully revised the self-citation references throughout the manuscript. To avoid redundancy and ensure objectivity, we have removed the majority of self-citations and retained only two references that are directly relevant and essential to support specific statements.

Reviewer 2 Report

Comments and Suggestions for Authors

The authors have addressed my concerns satisfactorily, and I have no additional comments.

Author Response

Reviewer #2:

The authors have addressed my concerns satisfactorily, and I have no additional comments.

RESPONSE: We sincerely appreciate your positive feedback and are grateful for your constructive comments during the review process. We are pleased to know that our revisions have satisfactorily addressed your concerns.

Reviewer 3 Report

Comments and Suggestions for Authors

The Fig 9, histopathology findings should be properly labelled with arrow marks.

Author Response

Reviewer #3:

The Fig 9, histopathology findings should be properly labelled with arrow marks.

RESPONSE: We thank the reviewer for the constructive suggestion. In the revised manuscript, Figure 9 has been updated with clear arrow marks and geometric symbols to highlight the key histopathological features (glomerulus, normal tubules, tubular necrosis, dilatation, casts, luminal debris, and interstitial edema). The figure legend has also been revised accordingly to describe these indicators.